# *USP27X* variants underlying X-linked intellectual disability disrupt protein function via distinct mechanisms

Intisar Koch[1],*, Maya Slovik[2,3,]*, Yuling Zhang[4], Bingyu Liu[4], Martin Rennie[5], Emily Konz[1], Benjamin Cogne[6,7], Muhannad Daana[8], Laura Davids[9], Illja J Diets[10], Nina B Gold[11,12], Alexander M Holtz[13], Bertrand Isidor[6,7], Hagar Mor-Shaked[2,3], Juanita Neira Fresneda[14], Karen Y Niederhoffer[15], Mathilde Nizon[6,7], Rolph Pfundt[10], MEH Simon[16], APA Stegmann[17], Maria J Guillen Sacoto[18], Marijke Wevers[10], Tahsin Stefan Barakat[19,20], Shira Yanovsky-Dagan[3], Boyko S Atanassov[21], Rachel Toth[22], Chengjiang Gao[4,]*, Francisco Bustos[1,23,]*, Tamar Harel[2,3,]*

Neurodevelopmental disorders with intellectual disability (ND/ID) are a heterogeneous group of diseases driving lifelong deficits in cognition and behavior with no definitive cure. X-linked intellectual disability disorder 105 (XLID105, #300984; OMIM) is a ND/ID driven by hemizygous variants in the *USP27X* gene encoding a protein deubiquitylase with a role in cell proliferation and neural development. Currently, only four genetically diagnosed individuals from two unrelated families have been described with limited clinical data. Furthermore, the mechanisms underlying the disorder are unknown. Here, we report 10 new XLID105 individuals from nine families and determine the impact of gene variants on USP27X protein function. Using a combination of clinical genetics, bioinformatics, biochemical, and cell biology approaches, we determined that XLID105 variants alter USP27X protein biology via distinct mechanisms including changes in developmentally relevant protein–protein interactions and deubiquitylating activity. Our data better define the phenotypic spectrum of XLID105 and suggest that XLID105 is driven by USP27X functional disruption. Understanding the pathogenic mechanisms of XLID105 variants will provide molecular insight into USP27X biology and may create the potential for therapy development.

## Introduction

Neurodevelopmental disorders with intellectual disability (ND/ID) are highly incapacitating conditions that affect 2–3% of the population (Leonard & Wen, 2002; Schalock et al, 2007; Maulik et al, 2011). Among these, ~15% are X-linked (Vissers et al, 2016) and over 160 genes on the X chromosome have been associated with X-linked intellectual disability (XLID) (Schwartz et al, 2023). Affected individuals have lifelong impairments in cognitive and adaptive functions that first manifest during early infancy. There is no cure for ND/ID, and currently available treatments are largely focused on symptom management (Picker & Walsh, 2013). An enhanced understanding of the molecular mechanisms underlying ND/ID may guide the potential development of therapeutic interventions.

Protein modification by ubiquitylation can drive changes in protein stability, protein–protein interaction, localization, and function in signaling pathways (Yau & Rape, 2016). Ubiquitylation is an essential cellular mechanism that is frequently disrupted in ND/ID (Ebstein et al, 2021). The mechanism involves the covalent transfer of the 76-aa protein ubiquitin to primarily lysine residues in protein substrates in a three-step enzymatic cascade (Hershko et al, 2000; Petroski, 2008), which requires ATP consumption and involves the activity of E1 activating, E2 conjugating, and E3 ligase enzymes (Pickart, 2001). Among the ubiquitylation genes associated with ND/ID, E3 ligases are the largest subgroup of which the

[1]Pediatrics and Rare Diseases Group, Sanford Research, Sioux Falls, SD, USA   [2]Faculty of Medicine, Hebrew University of Jerusalem, Jerusalem, Israel   [3]Department of Genetics, Hadassah Medical Center, Jerusalem, Israel   [4]Department of Immunology, School of Biomedical Sciences, Shandong University, Jinan, PR China   [5]School of Molecular Biosciences, College of Medical Veterinary and Life Sciences, University of Glasgow, Glasgow, UK   [6]Nantes Université, CHU de Nantes, CNRS, INSERM, L'institut du thorax, Nantes, France   [7]Nantes Université, CHU de Nantes, Service de Génétique médicale, Nantes, France   [8]Child Development Centers, Clalit Health Care Services, Jerusalem, Israel   [9]Department of Neurosciences, Children's Healthcare of Atlanta, Atlanta, GA, USA   [10]Department of Human Genetics, Radboud University Medical Center, Nijmegen, Netherlands   [11]Massachusetts General Hospital for Children, Boston, MA, USA   [12]Department of Pediatrics, Harvard Medical School, Boston, MA, USA   [13]Division of Genetics & Genomics, Department of Pediatrics, Boston Children's Hospital, and Harvard Medical School, Boston, MA, USA   [14]Department of Human Genetics, Emory University School of Medicine, Atlanta, GA, USA   [15]Department of Medical Genetics, University of Alberta, Edmonton, Canada   [16]Department of Genetics, University Medical Center Utrecht, Utrecht, Netherlands   [17]Department of Clinical Genetics, Maastricht University Medical Center, Maastricht, Netherlands   [18]GeneDx, Gaithersburg, MD, USA   [19]Department of Clinical Genetics, Erasmus MC University Medical Center, Rotterdam, Netherlands   [20]Discovery Unit, Department of Clinical Genetics, Erasmus MC University Medical Center, Rotterdam, Netherlands   [21]Department of Pharmacology and Therapeutics, Roswell Park Comprehensive Cancer Center, Buffalo, NY, USA   [22]MRC Protein Phosphorylation and Ubiquitylation Unit, School of Life Sciences, University of Dundee, Dundee, UK   [23]Department of Pediatrics, University of South Dakota, Sioux Falls, SD, USA

Correspondence: tamarhe@hadassah.org.il; Francisco.bustos@sanfordhealth.org
*Intisar Koch, Maya Slovik, Chengjiang Gao, Francisco Bustos, and Tamar Harel contributed equally to this work

**Life Science Alliance**

prototype UBE3A is associated with Angelman disease (105830; MIM) (Kishino et al, 1997; Matsuura et al, 1997). The phenotypes associated with this subgroup are dependent upon the substrates and signals downstream of the ligase enzyme activity. Most of these syndromes include developmental delay, intellectual disability, dysmorphic facial features, hypotonia, and seizures and may be accompanied by congenital anomalies (Ebstein et al, 2021).

Ubiquitylation is reversible by the action of the deubiquitylase (DUB) family of proteases (Komander et al, 2009). DUBs are highly represented among pathogenic variants causing ND/ID (Hu et al, 2016; Santiago-Sim et al, 2017; Johnson et al, 2020; Beck et al, 2021). However, the impact of these variants on catalytic activity and enzymatic function remains poorly understood.

Ubiquitin-specific protease 27 X-linked (USP27X) is a DUB that has been associated to important cellular functions such as cell proliferation and immune response. (Atanassov et al, 2016; Weber et al, 2016; Dong et al, 2018; Guo et al, 2019; Lambies et al, 2019; Seo et al, 2020; Alam et al, 2022; Dold et al, 2022). Furthermore, USP27X targets the developmental regulator HES1 to regulate neuronal differentiation (Kobayashi et al, 2015). Importantly, hemizygous variants in the *USP27X* gene have been recently linked to an ND/ID referred to by OMIM as XLID disorder 105 (XLID105, #300984; OMIM) (Hu et al, 2016; Kniffin, 2016). However, the mechanisms by which XLID105 *USP27X* variants affect USP27X function are currently unknown.

Here, we combine human genetics, bioinformatics, cell biology, and biochemical approaches to study the molecular basis of XLID105. This disorder is characterized by intellectual disability (ID), speech delay, and autistic features. Via functional studies focusing on USP27X protein biology, we describe potential pathogenic mechanisms of the different XLID105 *USP27X* variants and propose that USP27X functional disruption is a major pathogenic mechanism of XLID105.

# Results

## Clinical reports of individuals with *USP27X* variants

Detailed information on 10 individuals from nine families (Fig 1A and B) with *USP27X* variants is provided in Tables 1, S1, and S2. All affected individuals were male and ranged in age from ~3-36 yr. ID and/or speech delay were seen in all individuals with variable expressivity. Other neurodevelopmental issues, including autism spectrum disorder (6/10), attention deficit and hyperactivity disorder (7/10), anxiety (3/10), and behavioral or social–emotional problems (5/10) were prevalent within the cohort (Fig 1C). Six individuals had motor delay with two of them exhibiting gait abnormalities (wide-based or nonspecific unstable gait). One individual had febrile seizures and another had refractory epilepsy. Ophthalmological abnormalities included myopia, hypermetropia, strabismus, and astigmatism. Most individuals had a head circumference within the normal range, but two were reported with microcephaly. Dysmorphic features were recorded in several individuals, but seemed mostly nonspecific, although cupped/protruding ears and an elongated face with pointed chin were noted in several individuals. Other clinical manifestations seen in a single family, and therefore not necessarily within the phenotypic spectrum, included precocious puberty, neurosensory hearing loss, metopic craniosynostosis, severe feeding difficulties, and pigmentation abnormalities.

## Exome sequencing identifies hemizygous variants in *USP27X*

Variants in *USP27X* (NM_001145073.3) were identified in all individuals (Fig 1B). These included two stop-gain variants (c.106C>T; p.[Gln36Ter] and c.394G>T; p.[Glu132Ter]), one frameshift variant (c.1205dup; p.[Ala403fs]), and five missense variants (c.226G>A; [p.Gly76Ser], c.431A>G; p.[Tyr144Cys], c.541A>G; p.[Lys181Glu] c.937T>G; p.[Phe313Val], and c.1211G>A; p.[Ser404Asn]). The stop-gain would be expected to escape nonsense mediated decay, as *USP27X* has a single exon, yet would lead to premature truncation of the 438 aa protein. The missense variant p.(Ser404Asn) has one hemizygous call in gnomAD but was identified in two unrelated families with overlapping phenotypes and was therefore pursued. Similarly, a variant affecting the same residue as p.(Lys181Glu) - p.(Lys181Asn) was seen once as hemizygous in gnomAD. The other variants were neither observed in gnomAD nor in the TOPMed Bravo database. Computational prediction tools suggest either a deleterious effect or ambivalent predictions for the different variants (Table S3). CADD scores of the variants ranged from 21.9 to 25.5. All mothers available for testing were heterozygous for the variant. The mothers of two individuals were not available for testing (families 8 and 9). Whereas six mothers were reported to have normal cognitive function, two had mild ID: one with a frameshift variant and another with unknown *USP27X* variant status, but who had a child with a stop-gain variant. Clinical information for one biological mother was missing. This presentation is consistent with the X-linked recessive inheritance pattern, where males are affected, and females may have either no or mild manifestations.

Missense variants in *USP27X* found here and the previously described variant c.1141T>C; p. (Tyr381His) (Hu et al, 2016) result in changes to residues conserved across mammals (Fig S1). The fact that we observe similar clinical manifestations in individuals with missense and truncating *USP27X* variants suggests that missense variants likely result in the disruption of USP27X function and can affect one or more aspects of USP27X protein biology. Therefore, we decided to functionally characterize the missense variants, hereafter referred to by short nomenclature: G76S, Y144C, K181E, F313V, Y381H, and S404N.

## XLID105 USP27X mutants are stable and correctly localized to the nucleus

To investigate how *USP27X* variants affect USP27X protein localization and stability, we introduced human USP27X into $Usp27x^{-/y}$ mouse embryonic stem cells (ESCs) (Atanassov et al, 2016). WT USP27X is localized mainly to the nucleus with some cytoplasmic expression (Fig 2A) as previously described (Atanassov et al, 2016; Dold et al, 2022). Similar to the WT, XLID105 mutants largely localize to the nucleus (Fig 2A). Likewise, when we compared protein stability of WT and XLID105 mutant USP27X in a cycloheximide chase experiment, no significant differences were detected (Fig 2B). Taken

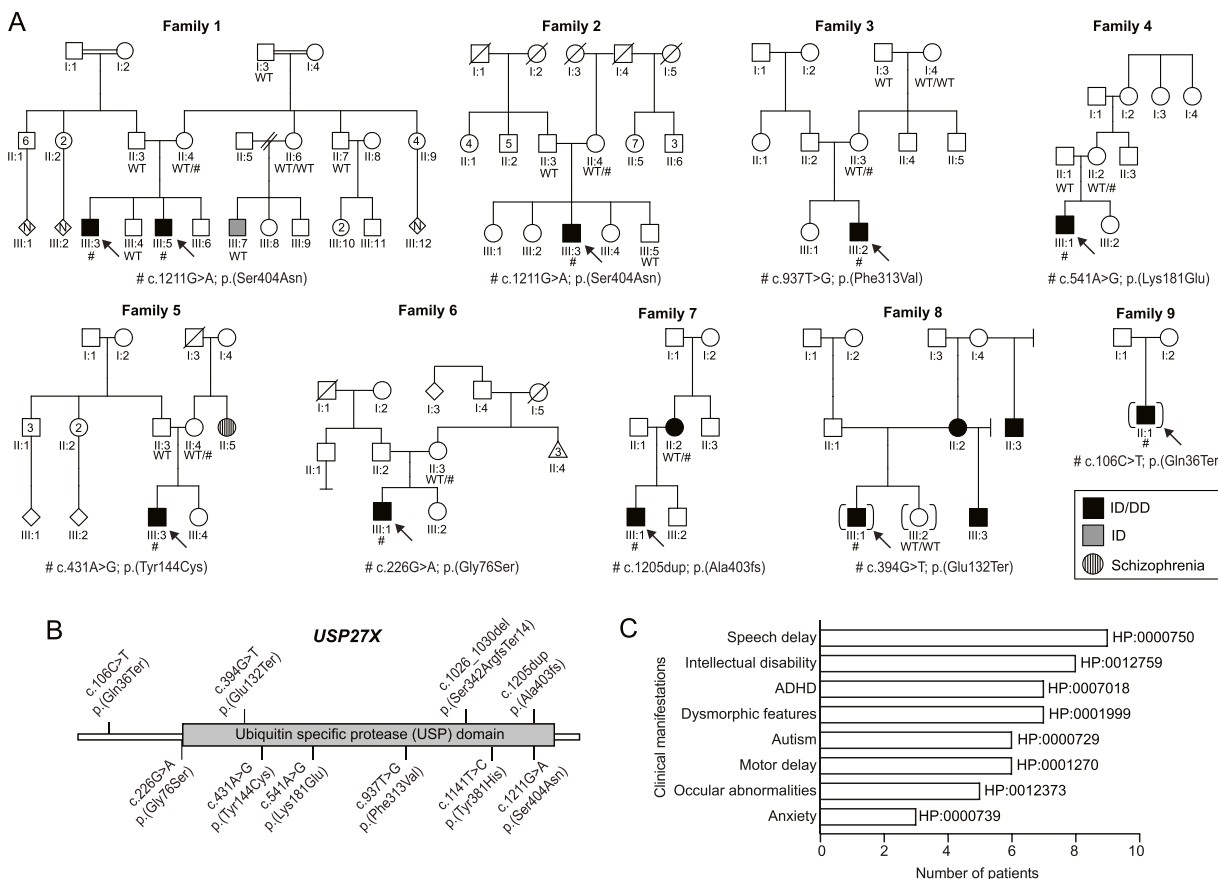

**Figure 1. Pedigrees suggestive of X-linked inheritance.**
**(A)** Pedigrees for variant segregation in nine families with 10 affected individuals harboring *USP27X* variants drawn using QuickPed (Vigeland, 2022). The medical history of family members is described in the supplementary material. **(B)** Location of variants on a 2D schematic diagram of the USP27X protein. **(C)** Bar graph depicting the prevalence of the most commonly shared clinical manifestations. Human Phenotype Ontology (HP) terms are shown.

together, these data indicate that these XLID105 variants do not have a major impact on USP27X localization or protein stability.

## XLID105 variants are predicted disruptive of the USP27X protein structure

Mutations in DUB protein sequence may disrupt its folding and, in turn, its function, (Hu et al, 2002) so we evaluated the structural relevance of USP27X residues mutated in XLID105. USP family enzymes are characterized by a USP domain that catalyzes ubiquitin removal (Hu et al, 2002, 2005; Renatus et al, 2006). This domain is formed by three subregions: the fingers, the palm, and the thumb (Hu et al, 2002). All three subregions accommodate ubiquitin, with the palm and thumb catalyzing ubiquitin cleavage. To gain insight into how XLID105 variants may impact USP27X structure and ubiquitin recognition, we analyzed the AlphaFold model of USP27X and computed a ColabFold model of USP27X bound to ubiquitin (Fig 3A and B). The modeled USP fold and ubiquitin were confidently predicted, as was the interaction between USP27X and ubiquitin (Fig S2A and B). In these models, we analyzed the location of residues in USP27X that are mutated in XLID105 (G76, Y144, K181, F313, Y381, and S404). K181 is adjacent to the thumb, whereas G76 and Y144 are

located within this subdomain of the USP fold. F313, Y381, and S404 are located within the palm subdomain (Fig 3A). The sidechain of S404 contributes intramolecular hydrogen bonds (Fig 3A). This structural feature is highly conserved in the closely related DUBs USP22 and USP51 where S503 in USP22 and T689 in USP51 stabilize a similar local structure (Fig S2C). Our model of USP27X bound to ubiquitin resembles a prototypical USP domain–ubiquitin interaction (Hu et al, 2002). Importantly, the sidechain of Y381 of USP27X forms hydrogen bonds with the backbone of ubiquitin, whereas F313 buttresses the ubiquitin tail (Fig 3B). Relative solvent accessible surface area (RSA) calculations indicate that Y144, K181, and S404 are relatively surface-exposed residues and that G76, F313, and Y381 are buried within the predicted structures (Fig 3C).

We predict that XLID105 variants F313V and Y381H disrupt the interaction with ubiquitin, whereas G76S and S404N may perturb the USP fold. In particular, the introduction of side chain atoms through G76S likely results in steric clashes with surrounding residues. The charge reversal of K181E may also have significant effects on the protein structure. However, from these predictions, the structural impact of the Y144C variant remains unclear suggesting that this variant affects other aspects of USP27X biology.

**Table 1. Detailed clinical information of affected individuals.**

| Individual | Family 1: III:3 | Family 1: III:5 | Family 2: III:3 | Family 3: III:2 | Family 4: III:1 | Family 5: III:3 | Family 6: III:1 | Family 7: III:1 | Family 8: III:1 | Family 9: III:1 | Hu et al (2016) | Hu et al (2016) |
|---|---|---|---|---|---|---|---|---|---|---|---|---|
| Age at last examination | 9 yr 8 mo | 5 yr 8 mo | 36 yr | 3 yr 11 mo | 11 yr 4 mo | 11 yr 6 mo | 2 yr 11 mo[a] | 9 yr | 9 yr | 12 yr | NA | NA |
| Gender | male | male | male | male | male | male | male | male | male | male | male | male |
| Preterm | no | no | no | NA | yes (35 wk) | no | no | no | no | NA | NA | NA |
| Weight (Z score) | 31.6 kg (+0.15) | 20 kg (+0.07) | 96 kg | 19.9 kg (+1.65) | 29 kg (−1.49) | 44.8 kg (+0.66) | (−1.37) | 27.2 kg (−0.38) | 24.1 kg (−1.27) | 35 kg (−0.81) | NA | NA |
| Height (Z score) | 143 cm (+1.04) | 114 cm (+0.27) | 165.5 cm (−1.54) | 105.8 cm (+1.06) | 140 cm (−0.65) | 154.9 cm (+1.30) | (+0) | 138.5 cm (+0.80) | 132 cm (−0.24) | 145 cm (−0.43) | NA | NA |
| Head circumference (Z score) | 53.3 cm (+0.46) | 52.5 cm (+0.84) | 56.2 cm (−0.67–0) | 52 cm (+1.19) | 50 cm (−2.44) | 56 cm (+1.97)[b] | (−0.7) | 49 cm (−2.74) | 54.5 cm (+1.45) | 53 cm (−0.48) | NA | NA |
| Head circumference (phenotype) | normal | normal | normal | normal | microcephaly | normal[b] | normal | microcephaly | normal | normal | NA | NA |
| Intellectual disability/Low developmental quotient | yes | yes | yes | yes | no | yes | yes | yes | yes | borderline | yes | yes |
| Speech delay | yes | yes | no | yes | yes | yes | yes | yes | yes | yes | yes | NA |
| Autism | yes | yes | no | NA | no | yes | yes | no | yes | yes | NA | NA |
| ADHD | yes | yes | no | NA | yes | no | yes | yes | yes | yes | NA | NA |
| Behavior characteristics | irritability | irritability, aggressiveness | underdeveloped social-emotional abilities | NA | no | calm, easy-going, seeks auditory input | no | happy, sometimes to extremes | emotional outburst, unable to regulate emotions | no | behavioral problems | NA |
| Mental illness | anxiety | no | no | NA | no | no | no | no | anxiety | anxiety | NA | NA |
| Motor delay | no | no | no | yes | no | yes | yes | yes | yes[c] | yes | NA | NA |
| Gait abnormality | no | no | no | NA | no | wide-based gait | no | unstable gait, clumsiness | no | no | NA | NA |
| Epilepsy | no | no | no | no | no | refractory[d] epilepsy | no | no | no | febrile seizures | NA | NA |
| Dysmorphic features | elongated face, elongated cupped ears, pointed chin | long eyelashes, fleshy earlobes, pointed chin, joint laxity | long and narrow face-mild, long nose with low columnella, cervicothoracal kyphosis | fleshy earlobes-mild, upturned nose, broad feet | no | triangular-shaped face with prominent chin, broad forehead, thick arched eyebrows with synophrys, deep-set eyes, down-slanting palpebral fissures, everted lower lids, gum hypertrophy, widely-spaced teeth, 5th digits clinodactyly | low-set ears, small cupped ears with overfolded helices, telecanthus, deep-set eyes, small mouth | protruding ears, full eyebrows, flat philtrum, thin upper lip vermillion, scapula alata, 5th digits clinodactyly | no | no | NA | NA |
| Hearing deficit | no | no | no | no | no | no | no | no | mild bilateral sensorineural hearing loss | no | NA | NA |

**Table 1. Continued**

| Individual | Family 1: III:3 | Family 1: III:5 | Family 2: III:3 | Family 3: III:2 | Family 4: III:1 | Family 5: III:3 | Family 6: III:1 | Family 7: III:1 | Family 8: III:1 | Family 9: III:1 | Hu et al (2016) | Hu et al (2016) |
|---|---|---|---|---|---|---|---|---|---|---|---|---|
| Ocular abnormalities | no | suspected myopia | no | strabismus | no | no | myopia, astigmatism, exotropia, strabismus | wears glasses, strabismus | no | hypermetropia | NA | NA |
| Other phenotypic abnormalities | no | no | no | no | severe feeding difficulties | precocious puberty | no | urine incontinence | no | metopic craniosynostosis, pigmentation anomaly | NA | NA |
| Imaging | NA | NA | NA | NA | NA | Brain MRI (2 yr): mild narrowing of posterior body of corpus callosum; Brain MRI (11 yr): normal; Brain PET/CT scan (11 yr): hypometabolism of parietotemporal regions, less severe involvement of frontal lobes, precuneus, basal ganglia, thalami, midbrain | Brain MRI: normal; Renal US: normal | NA | Echocardiogram: normal | NA | NA | NA |
| Mother's cognitive function | normal | normal | normal | normal | normal | normal | normal | mild ID | mild ID | NA | NA | NA |
| Other genetic evaluation[e] | FMR1: normal; CMA: no P/LP variants | FMR1: normal; CMA: normal | FMR1: normal; CMA: normal | FMR1: normal; CMA: normal | CMA: normal | Karyotype: 46,XY; CMA: no P/LP variants; RSS: normal methylation and UPD studies | CMA: no P/LP variants | no | FMR1: normal; CMA: normal; Hearing Loss Sequencing Panel | FMR1: normal; CMA: normal | NA | NA |
| Variant cDNA effect (NM_001145073.3) | c.1211G>A | c.1211G>A | c.1211G>A | c.937T>G | c.541A>G | c.431A>G | c.226G>A | c.1205dup | c.394G>T | c.106C>T | c.1026_1030del | c.1141T>C |
| Variant protein effect | p.(Ser404Asn) | p.(Ser404Asn) | p.(Ser404Asn) | p.(Phe313Val) | p.(Lys181Glu) | p.(Tyr144Cys) | p.(Gly76Ser) | p.(Ala403 fs) | p.(Glu132Ter) | p.(Gln36Ter) | p.(Ser342ArgfsTer14) | p.(Tyr381His) |
| Variant type | missense | missense | missense | missense | missense | missense | missense | frameshift | stop gain | stop gain | frameshift | missense |
| Variant inheritance | maternal | maternal | maternal | maternal (de novo in the mother) | maternal | maternal | maternal | maternal | NA | NA | maternal (not de novo in the mother) | NA |

[a]Growth parameters and dysmorphic features were evaluated at the age of 2 yr 11 mo. Other medical background is updated to current age of 8 yr 8 mo.

[b]Head circumference was measured at the age of 11 yr 1 mo. Parental head circumferences were not available.

[c]Fine, but not gross, motor delay.

[d]Up to six tonic-clonic seizures a day, at least twice a week, not amenable to several drugs including levetiracetam and clobazam.

[e]The detailed variants reported are described in the supplementary material.

Abbreviations: CMA, chromosomal microarray analysis; ID, intellectual disability; LP, likely pathogenic; NA, not available; P, pathogenic; RSS, Russell Silver syndrome; UPD, uniparental disomy.

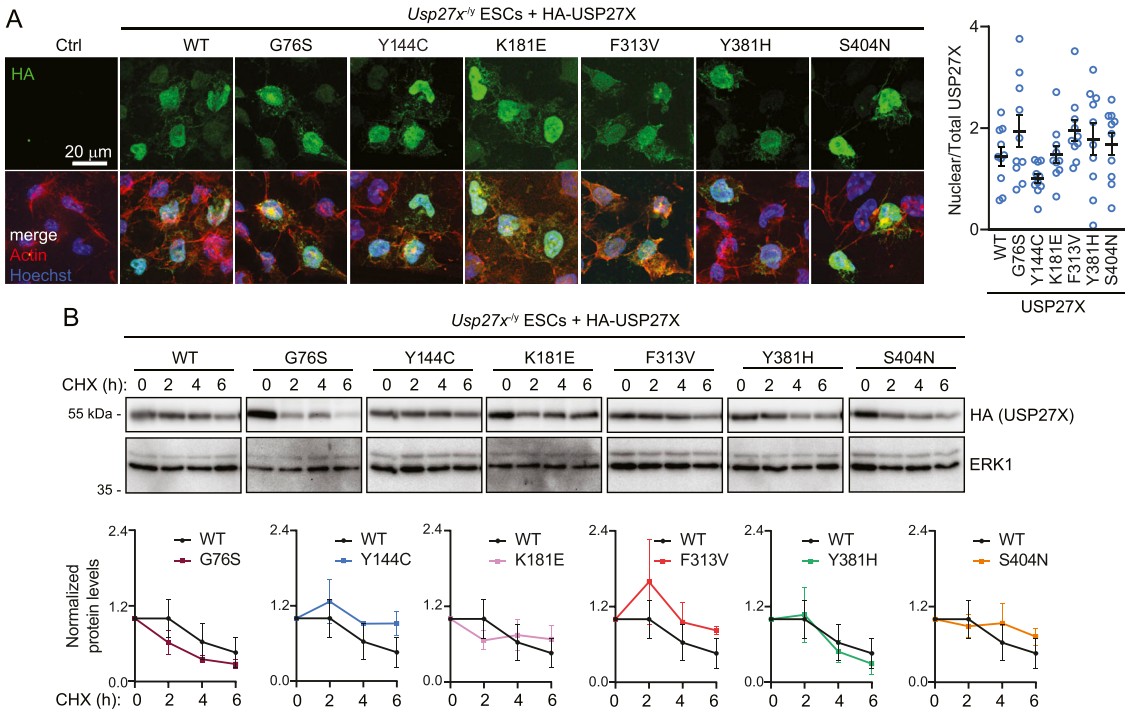

**Figure 2. XLID105 variants do not affect USP27X localization or protein stability.**
**(A)** Analysis of USP27X XLID105 mutant protein localization. *Usp27x⁻/y* ESCs were transfected with plasmids encoding the indicated HA-tagged USP27X mutants and their localization was analyzed via anti-HA (green) immunofluorescence and confocal microscopy. Actin staining (red) is shown as cytoskeleton marker and Hoechst (blue) was used as a nuclear marker. Scalebar: 20 μm, (n = 3). Quantification of the ratio between the nuclear and total HA-USP27X fluorescence intensity of 10 confocal image frames per variant is shown. No significant differences were found using one-way ANOVA analysis (data are presented as mean ± SEM). **(B)** Analysis of USP27X XLID105 mutant protein stability. *Usp27x⁻/y* ESCs were transfected with plasmids encoding the indicated HA-tagged USP27X constructs and cells were treated with cycloheximide for the indicated times. HA-tagged USP27X expression was analyzed via immunoblotting. ERK1 was used as a loading control. Quantification of relative HA-USP27X levels is displayed (data are presented as mean ± SEM, n = 3). No significant differences were found by t test analyses comparing each mutant with USP27X WT across four time points.
Source data are available for this figure.

## USP27X XLID105 variants display reduced catalytic activity

Given that deubiquitylating activity can vary depending on the substrate, we assessed the impact of XLID105 variants on the USP27X core catalytic mechanism using substrate-independent in vitro assays. We expressed recombinant wild-type or mutant USP27X and confirmed that the thermal stability of USP27X was not affected by the XLID105 variants in thermal shift assays (Fig S3A).

It has been reported that DUB pathogenic variants can selectively affect catalytic activity towards distinct ubiquitin chain linkages (Beck et al, 2021). Given that USP27X can cleave K48 and K63 ubiquitin chains (Ritorto et al, 2014), we directly measured the catalytic activity of the USP27X XLID105 variants on K48 and K63 di-ubiquitin chain cleavage assays in vitro. We found that the G76S, Y144C, F313V, Y381H, and S404N variants significantly impair catalytic activity, and observed similar results for both K48 and K63 di-ubiquitin (Fig 4A and B). These results are consistent with the predicted role of F313 and Y381 in ubiquitin interaction and the predicted structural relevance of G76S and S404N (Fig 3). These data indicate that most of the XLID105 missense variants disrupt USP27X catalytic activity, which may represent a major pathogenic mechanism in this disorder.

To further characterize these variants, we used a DUB activity-based probe. These probes consist of ubiquitin fused to a reactive carboxy terminal warhead. This warhead reacts with the DUB catalytic cysteine as this residue attacks the probe (Ekkebus et al, 2014). This assay allows for analysis of the critical steps of ubiquitin recognition and nucleophilic attack of the ubiquitin substrate isopeptide bond by the catalytic cysteine residue (Ekkebus et al, 2013). The ubiquitin–propargylamide (Ub-PRG) probe (Ekkebus et al, 2013) labels recombinant USP27X as it does USP2 in vitro (Fig S3B). As expected, USP27X Ub-PRG labelling was dependent on USP27X catalytic cysteine (C87) (Fig S3C) and was reduced when the reaction was performed in the presence of the broad DUB inhibitor PR-619 (Fig S3D). We next sought to determine the impact of XLID105 variants on USP27X ubiquitin recognition using the Ub-PRG probe labelling assay. We observed that the G76S, F313V, Y381H, and S404N mutant proteins displayed significantly decreased probe labelling compared with the control (Fig 4C). These data indicate that Y144C and K181E can still recognize ubiquitin and undergo nucleophilic attack, consistent with their surface exposure and location away from the active site (Fig 3); surprisingly, S404N cannot, suggesting the helix to which S404 forms a hydrogen bond (Fig S2C) may be allosterically important for ubiquitin binding and nucleophilic attack.

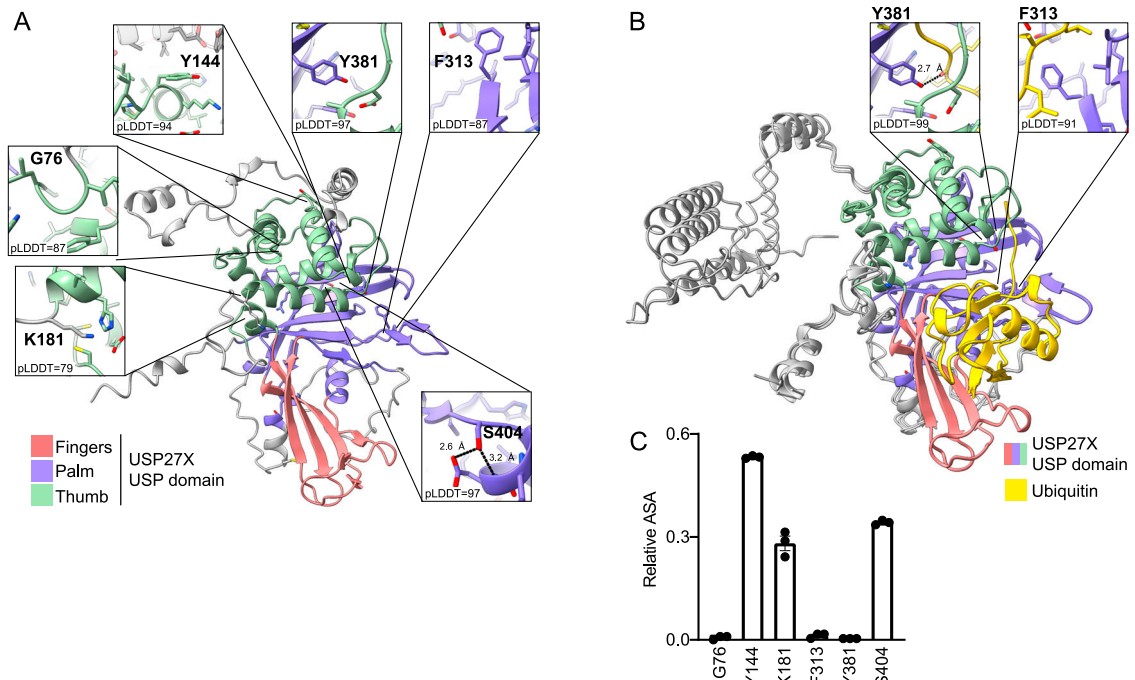

**Figure 3. Structurally relevant USP27X residues are mutated in XLID105.**
**(A)** USP27X predicted structure from the AlphaFold database. The fingers, palm, and thumb subdomains are colored pink, purple, and green, respectively. Insertions in the USP domain (>6 residue stretches with predicted Local Distance Difference Test scores <80) are colored gray. Residues found to be mutated in individuals with XLID105 are highlighted in boxes, with the confidence score (predicted Local Distance Difference Test) given. Hydrogen bonds are shown as dashed lines. **(B)** F313 and Y381 are predicted to be important to USP27X ubiquitin binding. The USP27X–ubiquitin complex predicted using ColabFold (three replicate computations shown). Close-up views of USP27X F313 and Y381 residues in contact with ubiquitin (yellow) are shown. Hydrogen bonds are depicted as dashed lines. **(B, C)** Analysis of fractional solvent accessible surface area of residues found to be mutated in XLID105 individuals, calculated from the AlphaFold structures from panel (B) (Data are presented as mean ± SEM).

## USP27X XLID105 variants alter interaction with a key USP27X protein partner

We also determined the impact of XLID105 variants on USP27X protein–protein interactions that are key for USP27X function. USP27X binds to a complex containing ATXN7L3 and ENY2 to catalyze histone H2B deubiquitylation and regulate cell proliferation (Atanassov et al, 2016). Correct function of this complex and histone H2B deubiquitylation are required for normal development (Herceg et al, 2001; Weake et al, 2008; Koutelou et al, 2019; Wang et al, 2021). Therefore, we sought to determine if USP27X XLID105 variants disrupted USP27X interaction with components of that complex. USP27X–ATXN7L3 interaction is detectable by co-immunoprecipitation when both proteins are expressed in *Usp27x*⁻/y ESCs (Fig 5A). When we compared the ATXN7L3 interaction levels of WT USP27X with the XLID105 variants (Fig 5B), we found that the G76S and Y381H variants significantly impair this interaction. We also found a significant increase in the enrichment of ATXN7L3 by USP27X Y144C. This evidence indicates that the G76S, Y144C, and Y381H variants alter a key USP27X protein interaction and could therefore affect developmentally relevant functions of USP27X.

## Discussion

Here, we expanded the phenotypic spectrum associated with *USP27X* variants by reporting an additional 10 individuals from nine families,

with different combinations of intellectual disability, developmental delay, autism spectrum disorder, ADHD, anxiety, and a tendency for ophthalmological abnormalities. We demonstrated that most XLID105 variants disrupt distinct aspects of USP27X protein biology that could perturb its function. Therefore, we propose that XLID105 pathogenesis is because of USP27X functional disruption. We determined that G76S, Y144C, and Y381H drive changes in protein–protein interactions and that five variants disrupt catalytic activity, with G76S, F313V, Y381H, and S404N being the most severe in their disruption (Table S3). This was consistent with our structural modelling which suggested that these four variants may disrupt the USP structure or interaction with ubiquitin. However, the K181E variant appeared to be unaffected in every test, aside from our in silico prediction of structural relevance. This variant may mediate disruption that is substrate or cell-context dependent. Otherwise, this may be a hypomorphic variant which correlates with the milder phenotype, that is, lack of intellectual disability and autism in the proband. Therefore, more functional studies or additional families with this variant are necessary to clarify its significance. Future studies may also allow us to decipher the genotype–phenotype correlation for this syndrome.

Several DUBs are associated with ND/ID syndromes (Jolly et al, 2022). However, structural studies and functional analyses have only recently been recognized as an important step to understanding pathogenesis (Ribarski et al, 2009; Beck et al, 2021; Chiang et al, 2021). Although members of the USP family are mutationally disrupted in ND/ID (Adorno et al, 2013; Hu et al, 2016; Chiang et al, 2021),

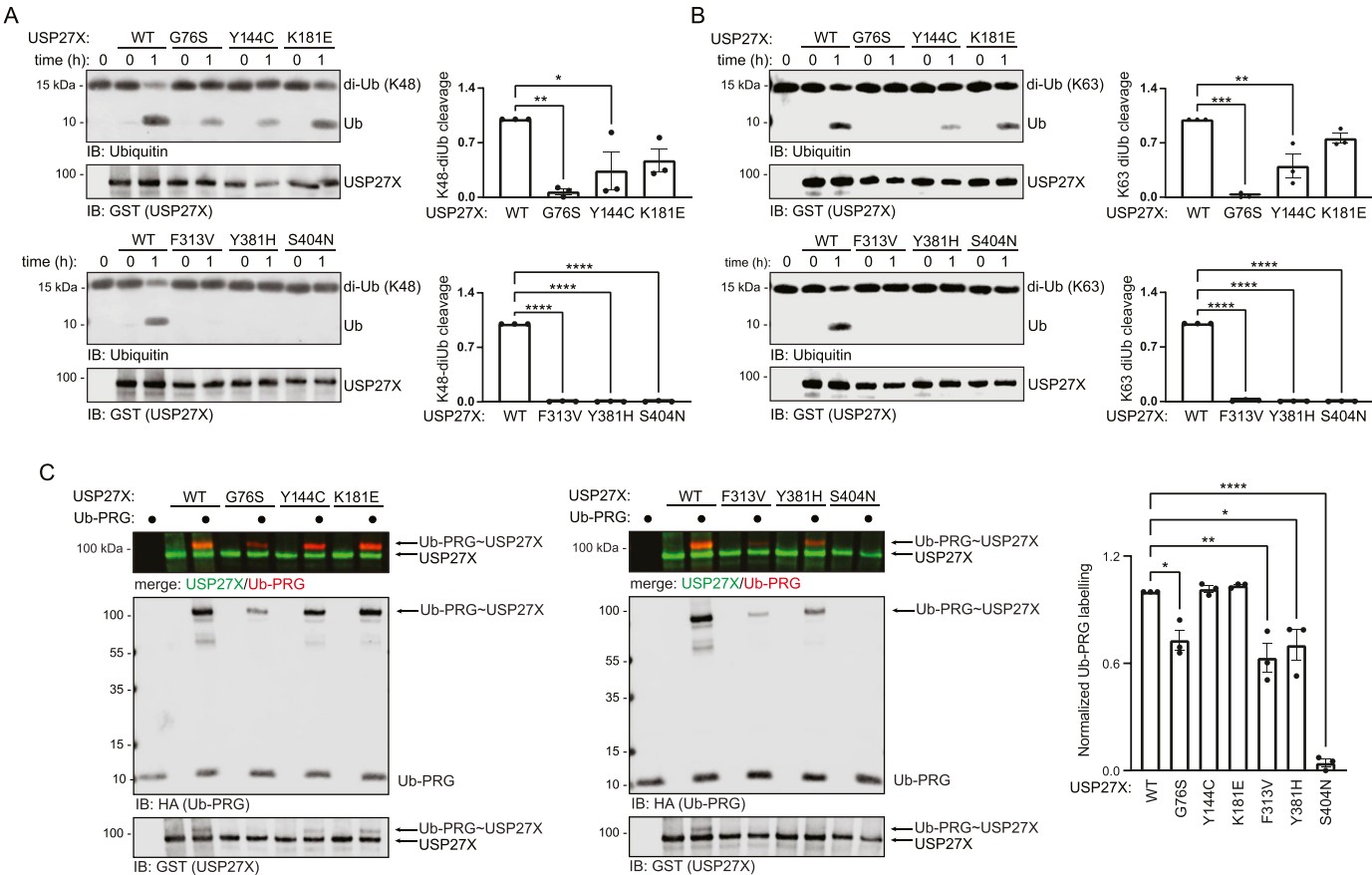

**Figure 4. XLID105 variants disrupt USP27X substrate-independent deubiquitylating activity.**
**(A, B)** GST-tagged WT or XLID105 mutant USP27X were incubated with K48 (A) or K63 (B) di-ubiquitin chains respectively. Data are presented as mean ± SEM, one-way ANOVA followed by Tukey's analysis, K48: P = 0.0474 (*), P = 0.0080 (**), P < 0.0001 (****); K63: P = 0.0041 (**), P = 0.0001 (***), P < 0.0001 (****). **(C)** GST-tagged wild-type or XLID105 mutant USP27X was incubated with Ub-PRG probe and labelling was analyzed via immunoblotting. Labelling quantification of three replicates is shown. Data are presented as mean ± SEM, one-way ANOVA followed by Tukey's analysis G76S: P = 0.0264(*), F313V: P = 0.0021 (**), Y381H: P = 0.0129 (*), and S404N: P < 0.0001 (****). A merged GST/HA (USP/Ub-PRG) image is displayed for visualization of the probe labelling.
Source data are available for this figure.

there is a lack of structural and direct functional studies of the impact of variants on USP DUB catalytic activity. We used AlphaFold (Jumper et al, 2021) to model the potential importance to folding of residues that are mutated in XLID105 and found that F313 and Y381 lie in regions within the USP fold that participate in ubiquitin recognition. We confirmed that the mutation of these residues in XLID105 disrupted function. Our results with the F313V variant are consistent with the fact that the equivalent residue of USP12 (F262) mutated to alanine dramatically reduces UAF1-stimulated catalysis of ubiquitin–AMC (Li et al, 2016). Although the USP domain in USP27X is consistent with the structure conserved within the USP family, it is known that DUB activity of individual members of the family can be modulated by specific allosteric regulation of this domain (Atanassov et al, 2016; Li et al, 2016; Rennie et al, 2021). Inserts in the USP domain can also contribute to specificity and regulation of USP27X catalytic activity (Tencer et al, 2016). Particularly for USP27X, an amino terminal extension resulting from a noncanonical translation start that occurs in some cell types (Atanassov et al, 2016; Dold et al, 2022) may play a role in regulation of catalytic activity. Furthermore, we performed in vitro assays using

recombinant proteins to assess the impact of variants on USP27X catalysis. Activity-based probes are a tool to rapidly visualize and measure changes in ubiquitin recognition and nucleophilic attack, whereas di-ubiquitin cleavage assays assess catalytic activity. Because these techniques measure different biochemical mechanisms, discrepancies in the results they yield are to be expected (Keijzer et al, 2023 Preprint). Using these assays, we demonstrated that the F313V and Y381H variants interfere with ubiquitin recognition or nucleophilic attack, most likely via an auxiliary mechanism stabilizing the ubiquitin tail. Most USP27X XLID105 variants tested disrupted catalytic activity in ubiquitin cleavage assays with differences in the severity of the disruption. This evidence supports the use of these approaches in the future to screen newly discovered USP27X or USP-type DUB variants for changes in activity that could drive intellectual disorders.

We demonstrated that the XLID105 variants G76S, Y144C, and Y381H alter USP27X interaction with the protein partner ATXN7L3. Y144 lies in a relatively exposed region within the palm subdomain of the USP fold. This result is compatible with a model in which Y144 either mediates protein–protein interaction or is subjected to

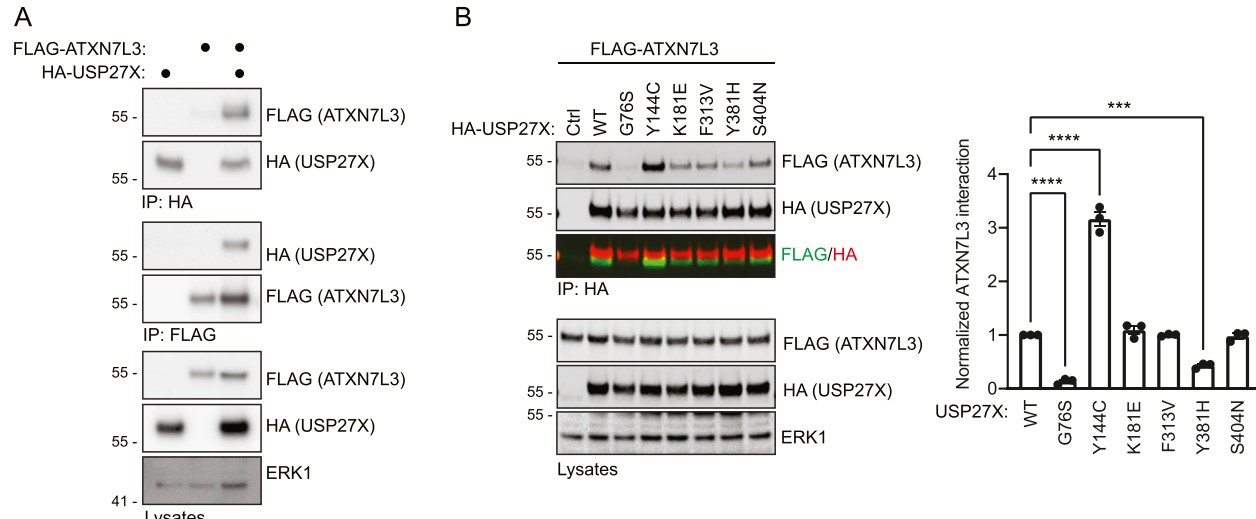

**Figure 5. USP27X XLID105 variants drive altered ATXN7L3 interaction.**
**(A)** Detection of specific USP27X–ATXN7L3 protein interaction. *Usp27x*$^{-/y}$ ESCs were transfected with plasmids encoding HA-tagged USP27X and FLAG-tagged ATXN7L3. Lysates were subjected to anti-HA or anti-FLAG immunoprecipitation and immunoblotting, (n = 3). **(B)** Analysis of ATXN7L3 interaction with USP27X XLID105 variants. **(A)** *Usp27x*$^{-/y}$ ESCs were transfected with plasmids encoding FLAG-tagged ATXN7L3 or HA-tagged WT or the indicated XLID105 mutants and samples were subjected to anti-HA immunoprecipitation as in (A), (n = 3). **(A, B)** Co-immunoprecipitation and lysate control immunoblots are shown in (A, B). **(A, B)** ERK1 levels are shown in the lysates as loading control for (A, B). **(B)** A merged HA/FLAG image is displayed for the immunoprecipitates in (B). Quantification of ATXN7L3 enrichment by the individual USP27X mutants is displayed. Data are presented as mean ± SEM, One-way ANOVA followed by Tukey's analysis P = 0.0003 (***), P > 0.0001 (****). Source data are available for this figure.

posttranslational modification. The Y144 change to cysteine in XLID105 is expected to dramatically affect this regulation. The impaired interaction of G76S and Y381H with ATXN7L3 may be associated with USP domain-folding defects. We have shown that ATXN7L3 interaction is important for USP27X-mediated histone H2B deubiquitylation at specific loci (Atanassov et al, 2016). ATXN7L3 is a limiting factor for the activity of distinct complexes that contain USP22 (SAGA), USP27X, or USP51, which mediate Histone H2B deubiquitylation on different target genes. These complexes compete for ATXN7L3 and are subjected to a fine balance (Atanassov et al, 2016). We predict that the variant-induced changes in USP27X affinity to ATXN7L3 may disrupt this balance. This could drive a dysregulation of the gene expression patterns that the ATXN7L3-containing complexes mediate.

USP27X is expressed in a wide range of tissues but is present at higher levels in the cortical and sub cortical brain regions and is even more elevated in the cerebellum (GTEx Consortium, 2013). Importantly, USP27X-mediated deubiquitylation of HES1 regulates neuronal differentiation (Kobayashi et al, 2015). Furthermore, USP27X mediates histone H2B deubiquitylation, which is critical for development (Weake et al, 2008). However, the function of USP27X in nervous system development is not fully understood. Determining the direct involvement of these USP27X-dependent axes in neurodevelopment will be key to better understanding the developmental basis of XLID105.

Our data establish that XLID105 is a USP27X functional disruption disorder. However, the idea of developing therapies for XLID105 raises the question of whether genetic intellectual disabilities are treatable disorders. Recent studies in animal models show encouraging results of the existence of a therapeutic window for intervention when individual ID genes are disrupted during development and reintroduced after birth (Mei et al, 2016; Creson et al, 2019; Terzic et al, 2021). It is yet to be determined whether this window exists for XLID105. If so, our study could set the basis for future therapies directed at restoring USP27X function.

# Materials and Methods

## Ethics statement

Families provided informed consent in accordance with the ethical standards of the responsible national and institutional committees on human subject research. Probands were ascertained through GeneMatcher (Sobreira et al, 2015).

## Exome sequencing

Exome analysis was pursued on DNA extracted from whole blood of the proband (families 1, 6, 7, 8, and 9) or proband and parents (families 2, 3, 4, and 5) at each contributing center. In Family 1, exonic sequences were enriched in the DNA sample using the IDT xGen Exome Research Panel V1.0 capture (Integrated DNA Technologies), and sequenced on a NovaSeq 6000 sequencing system (Illumina). Data analysis including read alignment and variant calling was performed with DNAnexus software (DNAnexus, Inc.) using default parameters, with the human genome assembly hg19/GRCh37 as reference. Variants were filtered out if they were off-target (intronic variants >8 bp from splice junction),

synonymous (unless <4 bp from the splice site) or had minor allele frequency > 0.01 in the Genome Aggregation Database (gnomAD) or local dataset. Bioinformatic predictions (i.e., PolyPhen, SIFT, MutationTaster, and/or CADD scores) were used to prioritize variants in genes with potential relevance to the phenotype.

## Clinical data

Percentiles for head circumferences were determined according to the Center for Disease Control and Prevention (CDC) for children and the Bushby et al charts for adults (Bushby et al, 1992).

## DNA cloning and mutagenesis

pCMV5 HA USP27X WT (MRC-PPU Reagents and Services DU36356) was generated by introducing the USP27X sequence (amplified by RT–PCR from human testis) in BamHI/NotI sites in pCMV5 HA1. HA-tagged USP27X point mutants were generated using the QuikChange II XL Site Directed Mutagenesis Kit (Agilent) or PCR and In-Fusion cloning (Takara Bio). HA USP27X C87A was generated at MRC-PPU Reagents and Services (DU36374). Mutagenized HA-tagged USP27X DNAs were then subcloned to pCAGGS puro (DU49023; MRC-PPU Reagents and Services) in an XhoI site or to pGEX6P1 (Cytiva) in a BamHI site via PCR and In-Fusion cloning. pCAGGS puro 3X FLAG ATXN7L3 was cloned by PCR and In-Fusion using pcDNA3.1+/C-(k)-ATXN7L3-DYK (Genscript OHu31133D Accession No: NM_020218.1) as a template to be cloned in a BamHI site in pCAGGS puro 3X FLAG (DU49056; MRC-PPU Reagents and Services). All constructs were confirmed using DNA sequencing.

## Protein expression and purification

pGEX6P1 USP2 and USP27X plasmids were transformed into BL21-CodonPlus (DE3)-RIPL or (New England Biolabs) Rosetta 2(DE3) Competent Cells (Novagen) via heat shock. Transformed colonies were grown in Terrific Broth media (GeneSee Scientific) containing 100 $\mu$g/ml ampicillin and 25 $\mu$g/ml chloramphenicol if required, at 37°C and 200$g$ in a MaxQ 4000 Benchtop Orbital Shaker (Thermo Fisher Scientific) up to OD600 = 0.5–0.6. Cultures were supplemented with 0.05 mM IPTG, then cooled down to 16°C, and incubated at 16°C and 200$g$ for 20 h. Bacterial pellets were obtained by centrifugation for 20 min at 4°C at 3,020$g$ and frozen overnight. Pellets were resuspended in MS500 lysis buffer (20 mM Tris pH 7.5, 300 mM NaCl, 0.5 mM TCEP, Lysozyme, and cOmplete Protease Inhibitor Cocktail Tablets [MilliporeSigma]). Bacterial lysates were sonicated in an analog Sonifier sonicator (Branson, Duty Cycle: 50, Output: 3, 2 min) and then centrifuged at 24,610$g$ at 4°C for 30 min. Cleared supernatants were transferred to columns containing glutathione agarose (Thermo Fisher Scientific) and passed through the column twice by gravity flow. Glutathione agarose-bound proteins were then washed three times with one resin volume of MS500 buffer (20 mM Tris pH 7.5, 500 mM NaCl, 0.5 mM TCEP). GST-tagged proteins were recovered by addition of elution buffer (MS500 buffer supplemented with 10 mM glutathione and 10 mM NaOH). Eluates were precipitated by addition of two volumes of 4M ammonium sulfate and mixing, followed by two centrifugation steps of 24,610$g$ at 4°C for 30 and 5 min, and stored at –80°C. Protein pellets were resuspended

in storage buffer (MS500 supplemented with 25% glycerol). Protein concentrations were measured using a NanoDrop spectrophotometer (Thermo Fisher Scientific), and protein quality and purity were analyzed via SDS–PAGE and Coomassie blue staining.

## Cell culture, transfection, and inhibitor treatments

*Usp27x$^{-/y}$* J1 mouse ESCs (Atanassov et al, 2016) were cultured in ES media (DMEM containing 10% FBS [vol/vol], 5% knockout serum replacement [vol/vol], 2 mM glutamine, 0.1 mM minimum essential media nonessential amino acids, 1 mM sodium pyruvate, penicillin, and streptomycin; 0.1 mM b-mercaptoethanol; and 100 g/ml LIF). Cells were grown in 0.1% gelatin (vol/vol) coated plates at 37°C and 5% $CO_2$. Plasmid transfections were performed using Lipofectamine LTX (Thermo Fisher Scientific) according to the manufacturer's instructions. 2–8 × 10$^4$ cells/cm$^2$ were combined with plasmid DNA and transfection reagents while in suspension and seeded in gelatin-coated plates for 48 h before analysis. For protein stability assays, cells were treated with 350 $\mu$M cycloheximide (MilliporeSigma–resuspended in DMSO).

## Immunoblotting

Cells were lysed in IP-MS buffer (20 mM Tris [pH 7.4], 150 mM NaCl, 1 mM EDTA, 1% Nonidet P-40 [NP-40] [vol/vol], 0.5% sodium deoxycholate [wt/vol], 10 mM b-glycerophosphate, 10 mM sodium pyrophosphate, 1 mM NaF, 2 mM Na3VO4, and cOmplete protease inhibitor cocktail tablets [MilliporeSigma]). Samples were cleared by centrifugation and protein concentration determined via the BCA assay (Thermo Fisher Scientific). 20–30 $\mu$g of protein were loaded in SDS–PAGE gels and transferred to PVDF or Nitrocellulose membranes. These were then blocked with TBS-Tween 20 (TBS-T) 5% nonfat milk (wt/vol) and incubated with primary antibodies. HRP or infrared dye-conjugated secondary antibodies were used for electrochemiluminescence or infrared detection using a ChemiDoc MP (Bio-Rad) or an Odyssey instrument (LI-COR). Figures were assembled using Image Lab (Bio-Rad), Image Studio (LI-COR), Inkscape 1.2 (Inkscape), and Illustrator (Adobe). Densitometric analyses were performed using ImageJ (NIH) or Image Studio (LI-COR) and data were analyzed in Excel (Microsoft) and GraphPad Prism 9 (GraphPad).

## Immunofluorescence

Cells were plated on 0.1% gelatin (wt/vol) coated glass coverslips and fixed using 4% paraformaldehyde (wt/vol) in PBS. Fixed cells were permeabilized with 0.5% Triton X-100 (vol/vol) in PBS and blocked with coverslip block buffer (4% Fish gelatin [wt/vol], 5.0% goat serum [vol/vol], and 1.2% BSA [wt/vol] in PBS). Cells were then incubated with an anti HA-tag antibody (1:10,000; BioLegend) followed by an Alexa 488-conjugated anti-mouse secondary antibody (1:500; Thermo Fisher Scientific) diluted in coverslip block buffer in a humid chamber. Actin red 555 reagent (Thermo Fisher Scientific) was added to the secondary antibody mix to stain actin. DNA was stained using Hoechst. Cells were mounted on glass slides using FluorSave (MilliporeSigma). Z-stack confocal images were acquired with a Nikon A1R confocal microscope using the NIS-Elements

software. Maximum intensity z-projections were generated using Image J (NIH) and images were assembled using Photoshop and Illustrator (Adobe). Quantification of the nuclear versus total HA-USP27X signal was performed using the adjust threshold and analyze particles tools of Image J (NIH).

### Co-immunoprecipitation

For co-immunoprecipitation assays, anti-HA Affinity Resin (Abcam) was washed with IPMS buffer and then blocked with 5% BSA (wt/vol) IPMS buffer for 1 h at 4°C. 0.3–0.5 mg of protein lysate was incubated overnight with 10 µl of blocked resin in a 500 µl total volume in a rotating wheel. Resin-bound immunoprecipitates were then separated by centrifugation and washed with IPMS buffer supplemented with 500 mM NaCl. Resin–protein complexes were then resuspended in 50% SDS–PAGE loading buffer (vol/vol) IPMS and boiled at 95°C for 5 min. Samples were loaded in SDS–PAGE gels and analyzed by immunoblotting as described above. ATXN7L3 interaction was displayed as the ratio between immunoprecipitated FLAG-ATXN7L3 and immunoprecipitated HA-USP27X. Data were normalized to the WT control.

### Protein sequence analysis and structure modelling

Protein alignments were performed using Clustal Ω (EMBL-EBI) and graphical representation was generated using ESPript (Robert & Gouet, 2014).

The predicted structure of human USP27X alone, USP22, and USP51 (UniProt accession: A6NNY8, Q9UPT9, and Q70EK9, respectively) were obtained from the AlphaFold Protein Structural Database (Jumper et al, 2021). Three structures of USP27X with ubiquitin were computed with AlphaFold-multimer (Evans et al, 2022 Preprint) using ColabFold (Mirdita et al, 2022) without templates and with amber relaxation and three recycles. ChimeraX (Pettersen et al, 2021) was used to prepare the figures.

The solvent accessible surface area of each residue of interest was normalized by the maximum allowed solvent accessibility for that residue type to give the relative solvent accessibility. The maximum allowed solvent accessibility for each residue type was obtained from theoretical estimates as determined in Tien et al, 2013 and the accessible surface area for each residue in each AlphaFold model was calculated using DSSP (Kabsch & Sander, 1983).

### In vitro activity assays

Thermal stability of recombinant proteins was estimated using thermal shift assays. 5 µM WT or mutant USP27X were incubated in DUB activation buffer (50 mM Tris–HCl pH 7.5, 50 mM NaCl, and 10 mM TCEP) with 5X SYPRO Orange (Thermo Fisher Scientific) in 25 µl reactions in a 96-well plate (Bio-Rad). After a 10-min incubation at RT, plates were transferred to a CFX96 real-time PCR system (Bio-Rad) and incubated at increasing temperatures from 10°C to 95°C in increments of 0.5°C for 10 s. Fluorescence emitted over time was measured (melt curve graph), and melt temperatures were estimated from the first derivative of fluorescence emission with respect to temperature (melt peak graph).

For di-ubiquitin cleavage assays, 2 µM WT or mutant USP27X were incubated in DUB activation buffer for 10 min at RT and mixed with 376 ng of K48 or K63 di-ubiquitin chains (South Bay Bio) for 1 h at 30°C. Reactions were stopped by adding SDS–PAGE loading buffer and samples analyzed via SDS–PAGE and immunoblotting. Di-ubiquitin cleavage was displayed as the levels of mono-ubiquitin in solution after the reaction. Data were normalized to the WT control.

For activity-based probe assays, 2 µM GST-USP2 or USP27X (WT or mutant) were incubated with 2 µM HA–ubiquitin–propargylmide (HA-Ub-PRG; South Bay Bio) in DUB activation buffer in a 10 µl reaction for 1 h at 30°C and 1,000g shaking in a Thermomixer F1.5 (Eppendorf). To inhibit DUB activity, PR-619 (Apex Bio) or DMSO (vehicle) was added to the reaction. Reactions were stopped by adding SDS–PAGE loading buffer and samples analyzed via SDS–PAGE and immunoblotting. Ub-PRG probe-labelling levels were displayed as the ratio between HA-Ub-PRG signal above USP27X and the total GST-USP27X signal. Data were normalized to the WT control.

## Data Availability

The ClinVar accession numbers for the DNA variant data are SCV003918884.1-SCV003918891.1. Reagents used or generated in this work listed in Table S4, materials, and raw data are available upon reasonable request from the corresponding authors. Plasmids generated by MRC-PPU Reagents and Services are available at http://mrcppureagents.dundee.ac.uk/.

## Supplementary Information

## Acknowledgements

The authors thank the patients and families who participated in this study. We thank Dr. Greg M. Findlay (University of Dundee) for critical reading of the article, and Dr. Matthew Schellenberg (Mayo Clinic, Rochester, M Nizon) for advice on protein purification protocols. This work was funded by the NIH grants R01CA272771 to BS Atanassov, and P30GM145398 to the Histology and Imaging core at Sanford Research. TS Barakat is supported by a ZonMw Vidi, Grant 09150172110002.

### Author Contributions

I Koch: data curation, formal analysis, visualization, and writing—original draft, review, and editing.
M Slovik: data curation, formal analysis, visualization, and writing—original draft, review, and editing.
Y Zhang: data curation, formal analysis, and writing—original draft, review, and editing.
B Liu: data curation, formal analysis, and writing—original draft, review, and editing.

M Rennie: data curation, formal analysis, and writing—original draft, review, and editing.

E Konz: data curation, visualization, and writing—review and editing.

B Cogne: data curation.

M Daana: data curation.

L Davids: data curation.

IJ Diets: data curation.

NB Gold: data curation.

AM Holtz: data curation.

B Isidor: data curation.

H Mor-Shaked: data curation.

J Neira Fresneda: data curation.

KY Niederhoffer: data curation.

M Nizon: data curation.

R Pfundt: data curation.

MEH Simon: data curation.

APA Stegmann: data curation.

MJ Guillen Saco: data curation.

M Wevers: data curation.

TS Barakat: data curation and writing—review and editing.

S Yanovsky-Dagan: data curation.

BS Atanassov: resources.

R Toth: resources.

C Gao: data curation, formal analysis, investigation, and writing—review and editing.

F Bustos: conceptualization, formal analysis, supervision, validation, investigation, and writing—original draft, review, and editing.

T Harel: conceptualization, data curation, formal analysis, supervision, investigation, visualization, and writing—original draft, review, and editing.

## Conflict of Interest Statement

MJ Guillen Sacoto is an employee of GeneDx, LLC. H Mor-Shaked is an employee of Geneyx Genomics. Other authors declare that they have no competing commercial interests.

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
