## [Reviewer comments · Life Science Alliance]

Life Science Alliance

USP27X variants underlying intellectual disability disrupt protein function via distinct mechanisms

Intisar Koch, Maya Slovik, Yuling Zhang, Bingyu Liu, Martin Rennie, Emily Konz, Benjamin Cogne, Muhannad Daana, Laura Davids, Illja Diets, Nina Gold, Alexander Holtz, Bertrand Isidor, Hagar Mor-Shaked, Juanita Neira Fresneda, Karen Niederhoffer, Mathilde Nizon, Rolph Pfundt, M.E.H. Simon, A.P.A. Stegmann, Maria Guillen Saco, Marijke Wevers, Tahsin Stefan Barakat, Shira Yanovsky-Dagan, Boyko Atanasov, Rachel Toth, Chengjiang Gao, Francisco Bustos, and Tamar Harel

DOI: <https://doi.org/10.26508/lsa.202302258>

Corresponding author(s): Tamar Harel, Hadassah Medical Center and Francisco Bustos, Sanford Research

Review Timeline:

Submission Date:	2023-07-07
Editorial Decision:	2023-08-10
Revision Received:	2023-11-23
Editorial Decision:	2023-12-15
Revision Received:	2023-12-22
Accepted:	2023-12-22

Transaction Report:

August 10, 2023

Re: Life Science Alliance manuscript #LSA-2023-02258

Prof. Tamar Harel
Hadassah Medical Center
Jerusalem 91120
Israel

Dear Dr. Harel,

Thank you for submitting your manuscript entitled "USP27X variants underlying intellectual disability disrupt protein function via distinct mechanisms" to Life Science Alliance. The manuscript was assessed by expert reviewers, whose comments are appended to this letter. We invite you to submit a revised manuscript addressing the Reviewer comments.

Thank you for this interesting contribution to Life Science Alliance. We are looking forward to receiving your revised manuscript.

Sincerely,

B. MANUSCRIPT ORGANIZATION AND FORMATTING:

Reviewer #1 (Comments to the Authors (Required)):

Individuals with neurodevelopmental disorders (ND) with intellectual disability (ID) exhibit a spectrum of phenotypes characterized by lifelong cognition and behavioral deficits. ND/ID has been found to be X-linked and understanding the molecular and biochemical mechanisms underlying disease progression of this disorder may lead to future therapeutic interventions.

Mutations in the deubiquitinating enzyme gene, USP27x, have been reported as causal for ND/ID. Koch and colleagues' work herein describe ten new cases of ND/ID owing to mutations in the USP27x gene. Their work reports XLID105 to be driven by hemizygote variants of USP27X gene. The authors utilized clinical genetics, bioinformatics, biochemical and cell biology approaches to define USP27X variants. They report that USP27x variants largely disrupt intrinsic deubiquitinating enzyme activity and alter protein-protein interactions. This reviewer finds the results potentially interesting, but several points need to be addressed to strengthen their findings prior to publication. The most significant points are listed below:

Major Points

1. Given that USP27X is reported to have DUB activity towards both K48-linked and K63-linked Ub chains, it remains undefined how the UPS27X mutant variants affect these ubiquitin linkages. Biochemical experiments to determine how these variants affect K48-link and K63-linked Ub chain hydrolysis can be empirically determined. Ideally, DUB assays on di- or tetra-ubiquitin chains can be done for each of the UPS27X variants. This analysis may also explain why some USP27X variants are able to deubiquitylate HES1 (Figure 5: Y144C, S404N) to some extent while others cannot (Figure 5: F313V, Y381H).
2. The authors report eight USP27X variants among 10 individuals but only four variants (Y144C, F313, Y381C and S404N) were functionally characterized. Moreover, of these four variants, Y381C is from a previous study reported in Hu et al., 2016. What are the molecular, cellular, and biochemical defects associated with the untested variants? All the other variants reported in this study should be functionally characterized to match the authors' assertion that they "determine the impact of gene variants on USP27X protein function."
3. Quantification of USP27X variant cellular localization (Figure 2A) is needed.
4. Figure 5 could be redone or at the very least quantified to reflect DUB activity more accurately. This is a noteworthy point given that the levels of USP27X variants vary across samples and these protein levels do not correlate with DUB activity.

Minor points

1. Figure 4C could be moved to supplementary figures since the inhibitor (PR-619) was not tested on any USP27X variant.
2. IP immunoblots figure labeling not consistent in both figures 5 and 6 with some figures lacking MW markers.
3. The activity of S404N variant of USP27X seen throughout the experiments opposes the claim that "S404N may compromise the USP" (figure 3B and 3C; line 188). More experiments needed to show exactly how S404N variant affects USP27X activity.

Reviewer #2 (Comments to the Authors (Required)):

In the present manuscript, Koch et al. identify 10 male individuals from 9 families with novel, hemizygous variants in USP27X, a gene that has previously been shown to be mutated in X-linked intellectual disability. The authors provide detailed clinical information on these patients, which exhibit a variety of neurodevelopmental and dysmorphic features. The identified variants follow an X-linked recessive inheritance pattern and result in protein truncations (3 variants) or amino acid substitutions (5 variants) in USP27X. Using a variety of cell biological, biochemical, and structural modeling approaches, the authors provide evidence for how 3 of the newly identified missense variants (p.Y114C, p.F313V, p.S404N) and the previously reported p.Y381H variant inactivate USP27X function. While none of the tested variants markedly affected protein stability or subcellular localization when ectopically expressed in Usp27x-/y mESCs, structural modeling identifies p.F313V and p.Y381H variants to likely disrupt interaction with ubiquitin and p.S404N to likely compromise USP fold. Consistent with these findings, the authors find the p.Y381H variant to be defective in reacting with a DUB activity probe in vitro. In addition, cellular deubiquitylation experiments are performed from which the authors conclude that p.F313V and p.Y381H are defective in deubiquitylation of the previously identified substrate HES1. Finally, co-immunoprecipitation experiments reveal that the p.Y114C variant increases interactions with the adaptor protein ATXN7L3.

This is a well-done study which identifies and clinical describes several novel patients with neurodevelopmental disease-causing

variants in USP27X, thus better defining the phenotypic spectrum of USP27X-deficient patients. The mechanistic analyses of the missense variants reveal diverse ways by which USP27X function is affected and suggest loss-of-function as the major mechanism underlying disease. These are important findings that will inform patient management and treatment and provide new avenues to study USP27X biology in the future.

Overall, the manuscript is well written, and most of the experiments are of high quality and support the author's claims. Aspects of this study that require strengthening are the in vitro and cellular deubiquitylation assays (see major points). In addition, I am raising some minor concerns, mostly regarding typos, inconsistencies, and clarifications that might help the reader to more readily understand the rationales for experiments.

Major points:

- The alpha-fold modeling studies suggest that p.F313V and p.Y381H variants likely disrupt interaction with ubiquitin and p.S404N likely compromises USP fold. However, in the in vitro assays in Figure 4, only p.Y381H shows a clear defect in reacting with the activity probe. This discrepancy should be addressed by the authors. E.g., since there is a large excess of HA-Ub-PRG over recombinant DUB in the assay, have the authors considered lowering/titrating the HA-Ub-PRG amount to reveal more subtle defects of the variants (in particular for F313V)? Similarly, have the authors considered a pre-treatment of the recombinant USP27X proteins at 30-37C before incubation with HA-Ub-PRG (which might reveal thermolability of protein structure for p.S404N)?
- Figure 5: I have several issues with the cellular deubiquitylation experiments and their interpretation. First, how do the authors know that the detected species are indeed ubiquitylated HES1 and not ubiquitylated proteins that are associated with HES1 upon MYC-IP? Could the authors do these IPs under denaturing conditions or perform denaturing His-Ub pull downs instead? It would also be important to include the molecular weight markers on the gels so the reader can appreciate where the Ub species are located on the blot relative to unmodified Hes1. Second, the anti-FLAG immunoblots of the inputs show that the expression levels of USP27X WT and mutant proteins are relatively uneven. From these assays, the authors claim that both p.F313V and p.Y381H are deficient removing ubiquitin from HES1, but how can the authors exclude that the increased signal they observe in the anti-HA blots of the MYC-IP fractions is not simply due to the lower expression of these variants compared to WT in the input fraction? I would recommend repeating these assays, using denaturing pull down/IP approaches and quantifying the loss of HES1 ubiquitylation relative to the amount of USP27X variant present in the input.

Minor points:

- The authors suggest that the increase interaction between USP27X p.Y144C and ATXN7L3 might be due to a disulfide bond (L240-244). Could the authors demonstrate this experimentally by co-expression of USP27X p.Y144C and ATXN7L3 WT and C84A followed by non-reducing SDS page and immunoblotting?
- One key conclusion is that the USP27X variants are loss-of-function mutations. However, it is not readily obvious how increases in the interaction of USP27X with its adaptor protein ATXN7L3 (which e.g. presumably brings USP27X to its targets) would result in loss of USP27X function. Could the authors more clearly discuss/speculate how they believe this would work? Is ENY2 binding also affected?
- Based on the alpha-Fold models shown in Figure 3, can the authors make any predictions how K181E and G76S would affect USP27X structure and/or interaction with ubiquitin?
- Line 153: Replace "Y381C" with "Y381H"
- Line 186: Replace "Figure 4A,C" with "Figure 3A,C"

Referee Cross-Commenting:

I agree with the comments of Reviewer #1

LSA 2023-02258 - Point by point assessment of reviewer's comments.

Reviewer #1 (Comments to the Authors (Required)):

Individuals with neurodevelopmental disorders (ND) with intellectual disability (ID) exhibit a spectrum of phenotypes characterized by lifelong cognition and behavioral deficits. ND/ID has been found to be X-linked and understanding the molecular and biochemical mechanisms underlying disease progression of this disorder may lead to future therapeutic interventions.

Mutations in the deubiquitinating enzyme gene, USP27x, have been reported as causal for ND/ID. Koch and colleagues' work herein describe ten new cases of ND/ID owing to mutations in the USP27x gene. Their work reports XLID105 to be driven by hemizygote variants of USP27X gene. The authors utilized clinical genetics, bioinformatics, biochemical and cell biology approaches to define USP27X variants. They report that USP27x variants largely disrupt intrinsic deubiquitinating enzyme activity and alter protein-protein interactions. This reviewer finds the results potentially interesting, but several points need to be addressed to strengthen their findings prior to publication. The most significant points are listed below:

We thank the reviewer for their positive comments about our manuscript and highlighting the importance of our findings.

Major Points

1. Given that USP27X is reported to have DUB activity towards both K48-linked and K63-linked Ub chains, it remains undefined how the UPS27X mutant variants affect these ubiquitin linkages. Biochemical experiments to determine how these variants affect K48-link and K63-linked Ub chain hydrolysis can be empirically determined. Ideally, DUB assays on di- or tetra-ubiquitin chains can be done for each of the UPS27X variants. This analysis may also explain why some USP27X variants are able to deubiquitylate HES1 (Figure 5: Y144C, S404N) to some extent while others cannot (Figure 5: F313V, Y381H).

We thank the reviewer for this important suggestion. We have now subjected the USP27X XLID105 variants to in vitro di-ubiquitin cleavage assays using K48 and K63 di-ubiquitin as substrates (Figure 4A, B and page 9, lines 6-15). We find exciting new data showing that most variants impair USP27X activity in these assays, with differences in the severity of the disruption. However, we observe similar results using either of these di-ubiquitin chains (Figure 4A, B and page 9, lines 6-15). This indicates that the variants affect catalytic activity independently of the ubiquitin linkage type.

2. The authors report eight USP27X variants among 10 individuals but only four variants (Y144C, F313, Y381C and S404N) were functionally characterized. Moreover, of these four variants, Y381C is from a previous study reported in Hu et al., 2016. What are the molecular, cellular, and biochemical defects associated with the untested variants? All the other variants reported in this study should be functionally characterized to match the authors' assertion that they "determine the impact of gene variants on USP27X protein function."

We appreciate the reviewer's concern, and we took this opportunity to assess these untested variants in our various assays. We find that the G76S and K181E variants are predicted to be critical for the Thumb subdomain folding (Figure 3 and page 8, line 7-9; 16-22). G76S shows disrupted catalytic activity in ubiquitin cleavage assays and probe labelling assays (Figure 4 and page 9-10). K181E decrease in activity was not statistically significant (Figure 4 and page 9-10). G76S impairs USP27X-ATXN7L3 interaction

(Figure 5B). This new data has provided important new functional information to better understand the molecular mechanisms disrupted by these variants. We also discussed about the phenotypic characteristics of the K181E variant that could correlate with the milder effects on functional assays (page 11, line 7-13)

3. Quantification of USP27X variant cellular localization (Figure 2A) is needed.

We thank the reviewer for this useful suggestion and have now included this information in Figure 2 and in the accompanying figure legend. The nuclear abundance of USP27X was not found significantly different in any of the variants tested. The high variability observed may be due to the experiment being performed as transient transfection.

4. Figure 5 could be redone or at the very least quantified to reflect DUB activity more accurately. This is a noteworthy point given that the levels of USP27X variants vary across samples and these protein levels do not correlate with DUB activity.

We appreciate the reviewer's concern. We decided to remove this figure because we were unable to observe USP27X effect on HES1 ubiquitylation using more direct ubiquitylation assays such as TUBE (tandem ubiquitin binding entities) pulldowns. Any changes observed were due to a global reduction on ubiquitin levels upon USP27X overexpression.

[Figure removed by editorial staff per authors' request]

We are currently uncertain about the functional relationship between USP27X and HES1. Due to the current absence of a high confidence USP27X substrate, we decided to focus our analysis on substrate independent in vitro DUB assays (Figure 4). Our future studies will aim to identify endogenous USP27X substrates and how they may be affected by USP27X variants.

Minor points

1. Figure 4C could be moved to supplementary figures since the inhibitor (PR-619) was not tested on any USP27X variant.

We thank the reviewer for this suggestion and have now moved this figure to the supplementary data (Figure S3D) along with its figure legend.

2. IP immunoblots figure labeling not consistent in both figures 5 and 6 with some figures lacking MW markers.

We thank the reviewer for pointing out this error. We have now included molecular weights in all immunoblots.

3. The activity of S404N variant of USP27X seen throughout the experiments opposes the claim that "S404N may compromise the USP" (figure 3B and 3C; line 188). More experiments needed to show exactly how S404N variant affects USP27X activity.

We appreciate the reviewer's concern; we have now subjected the S404N variant to in vitro di-ubiquitin cleavage assays using K48 and K63 di-ubiquitin as substrates. We find exciting new data showing that the S404N variant impairs USP27X activity in these assays (Figure 4 and pages 9-10). This data is consistent with our predictions on structural the role of the S404 residue. This result is surprising because S404 is a relatively exposed residue (Figure 3C), we suggest that the helix to which S404 forms a hydrogen bond (Figure S2C) may be allosterically important for ubiquitin binding and nucleophilic attack (page 10, line 4-6).

Reviewer #2 (Comments to the Authors (Required)):

In the present manuscript, Koch et al. identify 10 male individuals from 9 families with novel, hemizygous variants in USP27X, a gene that has previously been shown to be mutated in X-linked intellectual disability. The authors provide detailed clinical information on these patients, which exhibit a variety of neurodevelopmental and dysmorphic features. The identified variants follow an X-linked recessive inheritance pattern and result in protein truncations (3 variants) or amino acid substitutions (5 variants) in USP27X. Using a variety of cell biological, biochemical, and structural modeling approaches, the authors provide evidence for how 3 of the newly identified missense variants (p.Y114C, p.F313V, p.S404N) and the previously reported p.Y381H variant inactivate USP27X function. While none of the tested variants markedly affected protein stability or subcellular localization when ectopically expressed in Usp27x-/y mESCs, structural modeling identifies p.F313V and p.Y381H variants to likely disrupt interaction with ubiquitin and p.S404N to likely compromise USP fold. Consistent with these findings, the authors find the p.Y381H variant to be defective in reacting with a DUB activity probe in vitro. In addition, cellular deubiquitylation experiments are performed from which the authors conclude that p.F313V and p.Y381H are defective in deubiquitylation of the previously identified substrate HES1. Finally, co-immunoprecipitation experiments reveal that the p.Y114C variant increases interactions with the adaptor protein ATXN7L3.

This is a well-done study which identifies and clinical describes several novel patients with neurodevelopmental disease-causing variants in USP27X, thus better defining the phenotypic spectrum of USP27X-deficient patients. The mechanistic analyses of the missense variants reveal diverse ways by which USP27X function is affected and suggest loss-of-function as the major mechanism underlying

disease. These are important findings that will inform patient management and treatment and provide new avenues to study USP27X biology in the future.

Overall, the manuscript is well written, and most of the experiments are of high quality and support the author's claims. Aspects of this study that require strengthening are the in vitro and cellular deubiquitylation assays (see major points). In addition, I am raising some minor concerns, mostly regarding typos, inconsistencies, and clarifications that might help the reader to more readily understand the rationales for experiments.

We thank the reviewer for their enthusiastic comments about our manuscript and for highlighting the importance of our findings.

Major points:

- The alpha-fold modeling studies suggest that p.F313V and p.Y381H variants likely disrupt interaction with ubiquitin and p.S404N likely compromises USP fold. However, in the in vitro assays in Figure 4, only p.Y381H shows a clear defect in reacting with the activity probe. This discrepancy should be addressed by the authors. E.g., since there is a large excess of HA-Ub-PRG over recombinant DUB in the assay, have the authors considered lowering/titrating the HA-Ub-PRG amount to reveal more subtle defects of the variants (in particular for F313V)? Similarly, have the authors considered a pre-treatment of the recombinant USP27X proteins at 30-37C before incubation with HA-Ub-PRG (which might reveal thermolability of protein structure for p.S404N)?

This is an interesting suggestion made by the reviewer. In the original experiments, the extent of reaction for the PRG probe was low, and it would have been hard to titer this concentration down as this would have removed the labelling signal in the WT control. We believe that this was due to the low concentration of USP27X used in the assay, which may have also caused the variability observed within replicates. We addressed this problem by increasing the USP27X concentration in the assay, which has improved the signal obtained. The data now indicates that the G76S, F313V, and Y381H, and the S404N variants are impaired for probe labelling (Figure 4C, page 9 line 16-25 and page 10 line 1-6). This indicates that these variants perturb ubiquitin recognition or nucleophilic attack by the USP27X catalytic cysteine.

We directly tested the reviewer's concern about the thermal stability of the USP27X recombinant proteins via Thermal Shift Assays. We observed that the thermal stability of USP27X is not dramatically affected by the XLID105 variants (Figure S3A and page 9, line 3-5). Nevertheless, we propose that some of these variants affect the USP27X USP domain structure. We think that the variants may drive local effects on the fold that reduce catalysis without a dramatic effect on the melting temperature of the protein.

We have now clarified that the activity-based probe assay is addressing the initial step of deubiquitylation (page 9, line 18-20), which is ubiquitin recognition and nucleophilic attack, and we use this assay to gain more molecular insight on the mechanism perturbed by the variants. It has been shown that there are discrepancies between the activity-based probe assay and the ubiquitin cleavage assays as they measure different aspects of catalysis (discussed in page 12, lines 6-11). The ubiquitin cleavage assay is a more robust assay to measure DUB catalytic activity.

- Figure 5: I have several issues with the cellular deubiquitylation experiments and their interpretation. First, how do the authors know that the detected species are indeed ubiquitylated HES1 and not

ubiquitylated proteins that are associated with HES1 upon MYC-IP? Could the authors do these IPs under denaturing conditions or perform denaturing His-Ub pull downs instead? It would also be important to include the molecular weight markers on the gels so the reader can appreciate where the Ub species are located on the blot relative to unmodified HES1. Second, the anti-FLAG immunoblots of the inputs show that the expression levels of USP27X WT and mutant proteins are relatively uneven. From these assays, the authors claim that both p.F313V and p.Y381H are deficient removing ubiquitin from HES1, but how can the authors exclude that the increased signal they observe in the anti-HA blots of the MYC-IP fractions is not simply due to the lower expression of these variants compared to WT in the input fraction? I would recommend repeating these assays, using denaturing pull down/IP approaches and quantifying the loss of HES1 ubiquitylation relative to the amount of USP27X variant present in the input.

We appreciate the reviewer's concern. We decided to remove this figure because we were unable to observe USP27X effect on HES1 ubiquitylation using more direct ubiquitylation assays such as TUBE (tandem ubiquitin binding entities) pulldowns. Any changes observed were due to a global reduction on ubiquitin levels upon USP27X overexpression.

[Figure removed by editorial staff per authors' request]

We are currently uncertain about the functional relationship between USP27X and HES1. Due to the current absence of a high confidence USP27X substrate, we decided to focus our analysis on substrate independent in vitro DUB assays (Figure 4). Our future studies will aim to identify endogenous USP27X substrates and how they may be affected by USP27X variants.

Minor points:

- The authors suggest that the increase interaction between USP27X p.Y144C and ATXN7L3 might be due to a disulfide bond (L240-244). Could the authors demonstrate this experimentally by co-expression of USP27X p.Y144C and ATXN7L3 WT and C84A followed by non-reducing SDS page and immunoblotting?

We again thank the reviewer for another useful suggestion. We assessed whether USP27X Y144C and ATXN7L3 form a disulfide bond, as suggested by our USP27X-ATXN7L3 AlphaFold model. We analyzed the electrophoretic migration of USP27X WT or Y144C and ATXN7L3 WT or C84A alone or in combinations in *Usp27x*^{-/-} ESCs using Tris-Glycine non-reducing gels. We observed that although ATXN7L3 shows a retardation in its migration compared to the observed on reducing gels, this is not related to the presence of USP27X WT or Y144C. ATXN7L3 migration does not change with mutation of cysteine 84 to alanine. We also assessed whether cysteine 84 was relevant for the interaction between USP27X Y144C and ATXN7L3. Although we observe a decrease in the coimmunoprecipitation of USP27X Y144C and ATXN7L3, this change is not enough to suggest the existence of a disulfide bond between these proteins.

[Figure removed by editorial staff per authors' request]

We have now removed our AlphaFold prediction of USP27X-ATXN7L3 interaction from the figures along with the suggestion of the formation of a disulfide bond induced by the variant. Although the mechanism by which the Y144C variant increases the affinity of USP27X for ATXN7L3 is unknown and interesting, we believe that investigating that mechanism is beyond the scope of the current manuscript and will be explored in the future.

- One key conclusion is that the USP27X variants are loss-of-function mutations. However, it is not readily obvious how increases in the interaction of USP27X with its adaptor protein ATXN7L3 (which e.g. presumably brings USP27X to its targets) would result in loss of USP27X function. Could the authors more clearly discuss/speculate how they believe this would work? Is ENY2 binding also affected?

We appreciate the reviewer's concern. To avoid confusion, we have replaced the "loss of function" concept with "functional disruption" throughout the paper (page 3, line 14; page 5, line 15; page 11, line 3, and page 13, line 14). To explain why we think that an increased affinity for ATXN7L3 can drive functional disruption, we have included this new paragraph in the manuscript (page 12, line 25 to page 13, lines 1-5):

"ATXN7L3 is a limiting factor for the activity of distinct complexes that contain USP22 (SAGA), USP27X, or USP51, which mediate Histone H2B deubiquitylation on different target genes. These complexes compete for ATXN7L3 and are subjected to a fine balance (Atanassov et al, 2016). We predict that the variant-induced changes in USP27X affinity to ATXN7L3 may disrupt this balance. This could drive a dysregulation of the gene expression patterns that the ATXN7L3-containing complexes mediate."

Regarding ENY2, it has been shown that it directly binds to ATXN7L3 (PMID: 23591820, PMID: 27132940), and direct interaction between ENY2 and USP27X has not been demonstrated. Although exploring the impact of XLID105 variants on ENY2 interaction with the complex would be interesting, we believe that this is beyond the scope of the current manuscript.

- Based on the alpha-Fold models shown in Figure 3, can the authors make any predictions how K181E and G76S would affect USP27X structure and/or interaction with ubiquitin?

We appreciate the reviewer's concern, and we took this opportunity to assess these untested variants in our various assays. We find that the G76S and K181E variants are predicted to be critical for the Thumb subdomain folding (Figure 3 and page 8, line 7-9; 16-22). G76S shows disrupted catalytic activity in ubiquitin cleavage assays and probe labelling assays (Figure 4 and page 9-10). K181E decrease in activity was not statistically significant (Figure 4 and page 9-10). G76S impairs USP27X-ATXN7L3 interaction (Figure 5B). This new data has provided important new functional information to better understand the molecular mechanisms disrupted by these variants. We also discussed about the phenotypic characteristics of the K181E variant that could correlate with the milder effects on functional assays (page 11, line 7-13)

- Line 153: Replace "Y381C" with "Y381H"

We thank the reviewer for pointing out this error, which we have now rectified.

- Line 186: Replace "Figure 4A,C" with "Figure 3A,C"

We thank the reviewer for pointing out this error, which we have now rectified.

Referee Cross-Commenting:

I agree with the comments of Reviewer #1

December 15, 2023

RE: Life Science Alliance Manuscript #LSA-2023-02258R

Prof. Tamar Harel
Hadassah Medical Center
Jerusalem 91120
Israel

Dear Dr. Harel,

Thank you for submitting your revised manuscript entitled "USP27X variants underlying intellectual disability disrupt protein function via distinct mechanisms". We would be happy to publish your paper in Life Science Alliance pending final revisions necessary to meet our formatting guidelines.

- please add ORCID ID for the secondary corresponding author--they should have received instructions on how to do so
- please add the Twitter handle of your host institute/organization as well as your own or/and one of the authors in our system

A. FINAL FILES:

B. MANUSCRIPT ORGANIZATION AND FORMATTING:

The license to publish form must be signed before your manuscript can be sent to production. A link to the electronic license to publish form will be available to the corresponding author only. Please take a moment to check your funder requirements.

Sincerely,

Reviewer #1 (Comments to the Authors (Required)):

Individuals affected with neurodevelopmental disorders (ND) accompanied by intellectual disability (ID) manifest a nuanced spectrum of lifelong cognitive and behavioral challenges. The X-linked nature of ND/ID has been established, prompting an investigation to unravel the intricate molecular and biochemical mechanisms orchestrating the progression of this disorder, with an eye toward prospective therapeutic interventions.

Recent breakthroughs point to mutations in the USP27x deubiquitinating enzyme gene as culprits in ND/ID. In a significant stride forward, Koch and collaborators present a novel perspective, detailing ten new cases of ND/ID attributable to mutations in the USP27x gene. Their investigation singles out variants of the USP27X gene as the driving force behind XLID105. Employing a multifaceted methodology that incorporates clinical genetics, bioinformatics, and biochemical and cell biology techniques, the authors characterize USP27X variants. Their findings underscore that these variants significantly disrupt intrinsic deubiquitinating enzyme activity and induce alterations in protein-protein interactions.

The authors have adeptly responded to the concerns raised by the reviewer, fortifying their manuscript with additional data. This supplementary information enhances the robustness of their findings and contributes to the overall strength of the study.

Reviewer #2 (Comments to the Authors (Required)):

In the revised version, Koch et al, have provided additional experimental evidence to substantiate the claims of their original manuscript and have addressed all of my concerns.

December 22, 2023

RE: Life Science Alliance Manuscript #LSA-2023-02258RR

Prof. Tamar Harel
Hadassah Medical Center
Jerusalem 91120
Israel

Dear Dr. Harel,

Thank you for submitting your Research Article entitled "USP27X variants underlying intellectual disability disrupt protein function via distinct mechanisms". It is a pleasure to let you know that your manuscript is now accepted for publication in Life Science Alliance. Congratulations on this interesting work.

DISTRIBUTION OF MATERIALS:

Again, congratulations on a very nice paper. I hope you found the review process to be constructive and are pleased with how the manuscript was handled editorially. We look forward to future exciting submissions from your lab.

Sincerely,
